# Profound Nanoscale Structural and Biomechanical Changes in DNA Helix upon Treatment with Anthracycline Drugs

**DOI:** 10.3390/ijms21114142

**Published:** 2020-06-10

**Authors:** Aleksandra Kaczorowska, Weronika Lamperska, Kaja Frączkowska, Jan Masajada, Sławomir Drobczyński, Marta Sobas, Tomasz Wróbel, Kinga Chybicka, Radosław Tarkowski, Sebastian Kraszewski, Halina Podbielska, Wojciech Kałas, Marta Kopaczyńska

**Affiliations:** 1Department of Biomedical Engineering, Faculty of Fundamental Problems of Technology, Wroclaw University of Science and Technology, 27 Wybrzeze Wyspianskiego, 50-370 Wroclaw, Poland; aleksandra.kaczorowska@pwr.edu.pl (A.K.); kaja.fraczkowska@pwr.edu.pl (K.F.); sebastian.kraszewski@pwr.edu.pl (S.K.); halina.podbielska@pwr.edu.pl (H.P.); 2Department of Optics and Photonics, Faculty of Fundamental Problems of Technology, Wroclaw University of Science and Technology, 27 Wybrzeze Wyspianskiego, 50-370 Wroclaw, Poland; weronika.lamperska@pwr.edu.pl (W.L.); jan.masajada@pwr.edu.pl (J.M.); slawomir.drobczynski@pwr.edu.pl (S.D.); 3Department of Hematology, Blood Neoplasms and Bone Marrow Transplantation, Wroclaw Medical University, Pasteura 4, 50-367 Wroclaw, Poland; marta.sobas@umed.wroc.pl (M.S.); tomasz.wrobel@umed.wroc.pl (T.W.); 4Department of Experimental Oncology, Ludwik Hirszfeld Institute of Immunology and Experimental Therapy, Polish Academy of Sciences, Rudolfa Weigla 12, 53-114 Wroclaw, Poland; kinga.chybicka4@gmail.com (K.C.); wojciech.kalas@hirszfeld.pl (W.K.); 5Department of Surgical Oncology, Provincial Specialist Hospital, Iwaszkiewicza 5, 59-220 Legnica, Poland; rt@rakpiersi.net

**Keywords:** optical tweezers, DNA stiffness, cell nuclei stiffness, DNA structural changes

## Abstract

In our study, we describe the outcomes of the intercalation of different anthracycline antibiotics in double-stranded DNA at the nanoscale and single molecule level. Atomic force microscopy analysis revealed that intercalation results in significant elongation and thinning of dsDNA molecules. Additionally, using optical tweezers, we have shown that intercalation decreases the stiffness of DNA molecules, that results in greater susceptibility of dsDNA to break. Using DNA molecules with different GC/AT ratios, we checked whether anthracycline antibiotics show preference for GC-rich or AT-rich DNA fragments. We found that elongation, decrease in height and decrease in stiffness of dsDNA molecules was highest in GC-rich dsDNA, suggesting the preference of anthracycline antibiotics for GC pairs and GC-rich regions of DNA. This is important because such regions of genomes are enriched in DNA regulatory elements. By using three different anthracycline antibiotics, namely doxorubicin (DOX), epirubicin (EPI) and daunorubicin (DAU), we could compare their detrimental effects on DNA. Despite their analogical structure, anthracyclines differ in their effects on DNA molecules and GC-rich region preference. DOX had the strongest overall effect on the DNA topology, causing the largest elongation and decrease in height. On the other hand, EPI has the lowest preference for GC-rich dsDNA. Moreover, we demonstrated that the nanoscale perturbations in dsDNA topology are reflected by changes in the microscale properties of the cell, as even short exposition to doxorubicin resulted in an increase in nuclei stiffness, which can be due to aberration of the chromatin organization, upon intercalation of doxorubicin molecules.

## 1. Introduction

Anthracyclines are antibiotics produced by the bacterium *Streptomyces peucetius,* a species used to treat various cancers [1,2]. Most commonly used anthracyclines are daunorubicin (DAU) and its synthetic derivatives doxorubicin (DOX), epirubicin (EPI) and idarubicin (IDA). All these drugs have a wide mechanism of action and pass cellular and cytoplasmic membranes by diffusion. While in the cytoplasm, the anthracyclines affect ceramide metabolism [3,4,5] and the function of proteasome [2,6] that both lead to the induction of programmed cell death. The main mechanism of action of anthracyclines is related to their interaction with DNA [6,7]. They pass nuclear membrane using proteasome [7] and bind to DNA by covalent (alkylation, cross-linkages) and non-covalent bonds (intercalation), binding in a minor or major groove of a DNA double helix and through electrostatic interactions [8]. Most of the nuclear activity of anthracycline antibiotics is attributed to their ability to inhibit topoisomerase II (TOPO II) via formation of a ternary complex, which consists of DNA, TOPO II and a drug. The planar chromophores of the drug are involved in the DNA intercalation, whereas daunosamine is involved in the stabilization of the DNA complex with topoisomerase II [9]. It is worth to mention that intercalation itself, without trapping TOPO II, can lead to apoptosis. Changes in DNA and chromatin structure alter the function of nuclear factors and signaling pathways, which in consequence, leads to cell death [10].

DNA intercalation is the insertion of a flat drug between adjacent base pairs with the formation of at least one hydrogen bond. Under normal conditions, in a B-DNA double helix, the twist angle between adjacent base pairs in a B-DNA double helix is approximately 36°. After the formation of the intercalation complex, this angle is usually reduced by 10° to 26°. Moreover, the structure of the intercalated dsDNA relaxes, uncoils and lengthens by about 0.34 nm for every molecule that intercalated into the DNA [11].

Drugs intercalate DNA without specificity to any particular nucleotide sequence. However, it has been demonstrated that anthracycline antibiotics may have higher affinity to fragments rich in GC pairs [9,12,13]. There are studies that describe the intercalation of anthracyclines to DNA molecules, but according to our best knowledge, a comparison of different anthracyclines or their preferences for DNA of various GC content have never been investigated, especially at the nanoscale and single molecule level.

Such nanoscale conformational changes of a DNA helix can be studied by atomic force microscopy (AFM) [14]. By analyzing structural changes, especially in the length or orientation of nucleic acid strands before and after the drug interaction, it can be distinguished whether it interacts with DNA as a result of intercalation, alkylation, binding within the major or minor double helix groove, or a combination of these mechanisms [15,16]. As we have shown previously, also a damaging effect on the nucleic acid strand can be visualized and detected by AFM [17].

Structural changes in dsDNA affect its mechanical properties that are crucial for maintaining chromatin integrity. In this regard, holographic optical tweezers (OT) enable nanoscale studies of biological objects, e.g., examination of mechanical properties and structural changes in DNA complexes with proteins or drug molecules [18,19]. Due to the non-invasive and non-destructing character of such measurements, optical tweezers have become a valuable instrument that can be used to study biomechanical properties of DNA [20,21,22,23,24]. The sequence and structural selectivity of 15 different DNA binding agents was presented using a competition dialysis procedure. Different nucleic acid structures were dialyzed against ethidium, daunorubicin and actinomycin D, a common ligand solution [25].

Despite wide use of anthracyclines in the treatment of cancer, the data regard detailed structural changes in single DNA strands upon exposure to anthracycline antibiotics. For example, changes in the thermodynamic and mechanical profiles of a DNA strand due to the presence of antineoplastic drugs like PD153035 and cisplatin were demonstrated by using optical tweezers, but it was never presented for daunorubicin and epirubicin [26,27].

In this study, we used atomic force microscopy to visualize and measure nanostructural changes upon intercalation of DAU or DOX with dsDNA. Additionally, we adapted the technique of using microbeads for the quasi-static stretching of DNA strands treated with DAU or DOX anthracycline antibiotics. To determine whether the observed effects are related to the DNA composition, DNA molecules with three different GC/AT ratios were used.

## 2. Results and Discussion

### 2.1. In Silico Analysis Reveals Diverse Modes of Interactions of DNA with Anthracyclines

First, we performed numerical modelling of the interaction of DOX, DAU or EPI. In silico analysis revealed diverse modes of interaction of anthracycline antibiotics with DNA depending on the GC/AT ratio (Figure 1). Apart from the intercalation, binding to minor or major groove DNA should be expected. Interestingly, the later interactions can be quite common. The predicted elongation of dsDNA molecules was also calculated. The use of DNA strands differing in the percentage of GC to AT base pairs was intended to examine whether anthracyclines tend to bind to specific DNA double helix sites—between the GC nucleotides. The strongest lengthening of a DNA strand was anticipated for DAU on GC-rich dsDNA (70% of GC pairs) and AT-rich dsDNA (30% of GC pairs). The lengthening of balanced dsDNA (50% of GC pairs) was the weakest, suggesting the prevalence of modes of interaction other than intercalation. Nevertheless, greater elongation upon interaction with DOX was predicted for balanced dsDNA and GC-rich DNA. The presence of EPI may cause a substantial elongation of AT-rich dsDNA, about 30 Å, while the elongation with other DNA strands was weaker. A similar weak impact on the length of dsDNA, about 10 Å, was observed for DOX, regardless of the GC/AT ratio in the dsDNA.

It is worth to underline, that despite the high structural similarity of the drugs, in silico analysis revealed that modes of direct interactions of particular anthracyclines vary. This may influence the biological activity of particular anthracyclines.

### 2.2. Anthracyclines Increase Length and Decrease Height of dsDNA Molecule

To test different modes of interaction, revealed by in silico analysis, we performed direct studies of the topography of DNA molecules using AFM. Similar to the simulation conditions, we used DNA strands with three different GC/AT ratios (35%, 52% and 77% of GC), and approximately 500 bp long. The dsDNA was treated with DOX, DAU and EPI in low, equal 0.3 μM and high equal 2 μM concentrations. Intercalation involves inserting a molecule between two adjacent base pairs and the number of potential intercalation sites is one site less than the number of nucleotide pairs in the DNA strand. The concentration of 0.3 μM provides a number of sufficient anthracycline molecules to intercalate about 20% of the potential sites in the entire DNA molecule (1/5 drug/total DNA bp ratio). The concentration of 2 μM provides an excess amount of drug molecules relative to the potential sites of intercalation (4/3 drug/total DNA bp ratio). For each sample, 30 images were recorded and analyzed. The length and height of the DNA strands were determined using the Nanoscope 6.13.1.0 program. The height of DNA was defined as the difference between the highest point on the DNA strand and the mica foundation, while the length of DNA was defined as the length of the DNA contour between its two ends.

The initial size of dsDNA molecules measured in the control samples was constant, regardless of the GC/AT ratios: 144.76 ± 10.30 nm length and 1.044 ± 0.104 nm height for GC-rich dsDNA, 147.81 ± 6.80 nm length/1.013 ± 0.071 nm height for balanced dsDNA and 147.57 ± 7.48 nm length/1.012 ± 0.068 nm height for AT-rich dsDNA (Figure 1 and Figure 2).

We observed that treatment with all three anthracycline antibiotics DOX, DAU and EPI caused significant elongation and reduction in the height of the DNA molecules (Figure 2). Similar lengthening of the DNA molecules was also observed by AFM upon intercalation of YOYO-1 dye [28]. Interestingly, the increment of a DNA contour can be observed upon treatment with PD153035 in OT studies [26]. We found a strong correlation between the lengthening and tightening of the DNA and it was later confirmed by correlation analysis on GC-rich dsDNA (Appendix A).

DOX had the strongest overall effect on the DNA topology, causing elongation up to 191.48 ± 16.06 nm and tightening to only 0.489 ± 0.046 nm in a high concentration on GC-rich dsDNA. On the other hand, DAU had the minimum effect on the DNA topology. The observed effects were concentration-dependent. Interestingly, anthracyclines in a low concentration had a stronger effect on the DNA height than its length, while increasing the concentration of DOX or EPI, but not DAU, resulted in further significant elongation of the DNA molecules. Further, at low concentrations, all drugs had the lowest effect on AT-rich dsDNA and the highest effect on GC-rich dsDNA. Drugs did not differ greatly in their effect on GC-rich dsDNA at low concentrations. At high concentrations, the maximal effect was also observed on GC-rich dsDNA, but its extent differed greatly between the drugs. Increasing the concentration of DOX boosts its effect on both the length and height of the DNA molecules. This is unlike in the case of DAU, where increasing concentrations have a very limited effect on the DNA topology. Interestingly, in the case of AT-rich and balanced dsDNA, the effect on the DNA height dominated over elongation of the DNA, while a very strong effect on the DNA length was observed in GC-rich dsDNA. We observed that DOX and DAU seem to have a high preference for GC-rich dsDNA as it is significantly more influenced by these drugs, on the other hand. EPI had a similar effect on GC-rich and balanced dsDNA.

This allows us to draw general conclusions that (1) anthracyclines differ in their effect on the DNA molecules and (2) that this effect is stronger on GC-rich dsDNA.

### 2.3. Anthracyclines Induce Substantial Decrease of dsDNA Stiffness

For studies of DNA biomechanics and to find-out how anthracyclines change the mechanical properties of dsDNA, we adapted the OT system. During the analysis, one bead is held fixed (i.e., it does not move), while the second bead is pulled away. The scheme of the experimental set up is shown in Figure 3, while Figure 4 shows a typical record of the stiffness testing experiment. The stretching is performed in a quasi-static manner. The second bead is pulled away from the first one in a slow, stepwise way in order to reduce the negative influence of the drag from the surrounding medium (for a detailed discussion on the quasi-static method see Section 3.3.2). As the beads move further and further away from each other, the DNA chain gets untangled and straightened to its actual length, dictated by the number of base pairs in the chain. Then, the stretching begins. The forced elongation of the DNA molecule is initially proportional to the applied force and the slope of the fitted linear function (red line). At some point, the DNA strain reaches its limit and breaks. This is a very characteristic moment and can be easily observed under the microscope during the measurement and later during the analysis of the beads’ trajectories. The stiffness of the DNA chain is calculated as a slope of the linear function fitted to the curve and expressed in the unit pN/µm (picoNewton per micrometer). The linear fitting was performed using the standard linear regression in Curve Fitting Toolbox, MATLAB (Mathworks). According to our knowledge, the breaking force decrease may be caused by the experimental technique. A similar experimental setup was applied in studies of the biomechanical properties of phage lambda DNA in the presence of PD153035 and cisplatin [26,27]. In our case, the impact of anthracycline antibiotics on the biomechanics of the DNA molecules was studied using 4000 bp dsDNA (1.36 µm): AT-rich, balanced and GC-rich. The ratio of drug to DNA was selected to provide a small excess amount of drug molecules relative to the potential sites of intercalation.

The stiffness of untreated DNA varied between 21 to 24 pN/µm and it seems not to depend on the DNA composition. All examined drugs decreased the stiffness of dsDNA. This is consistent with previous observations [27], where the addition of another DNA-interacting drug, cisplatin, resulted in a decrease in the persistent length with an increased concentration of the drug. Interestingly, cisplatin does not primarily intercalate to DNA, suggesting that all modes of interaction can substantially affect the homeostasis of the dsDNA molecule. The greatest decrease in stiffness was observed for GC-rich dsDNA. In the case of DOX, a strong negative correlation with the GC pair content can be noticed. This drug reduced the stiffness of GC-rich dsDNA by 63%. On the other hand, the effect of DAU and EPI on stiffness was equally high for AT-rich and GC-rich dsDNAs. Thus, we conclude that DOX shows a tendency to intercalate in regions rich in GC pairs rather than AT pairs. Such a tendency is not observed for DAU and EPI.

In the same experimental setup, we measured the force needed to break down the studied molecules of dsDNA. In our experimental setup, we could not break untreated AT-rich dsDNA, as well as treated with DOX or DAU (Table 1). Upon treatment with EPI, dsDNA can be broken by the force of 8.5 pN. The anthracyclines do not have an impact on balanced dsDNA, but visibly reduce the mean force needed to break GC-rich dsDNA. Both analyses show a deteriorating impact of anthracyclines on dsDNA and indicate that GC-rich regions of DNA can be most vulnerable. This is important because such regions of genomes are enriched in DNA regulatory elements.

### 2.4. Biophysical Changes Are Reflected by Changes of Nuclei Size and Stiffness

In previous experiments, we demonstrated a number of nanoscale disturbances of dsDNA properties caused by anthracycline antibiotics. It has been suggested that the superhelical twist and higher order structures can prevent the DNA intercalation, and thus can serve as a natural barrier against some molecules [29]. It is always the question if and how these effects could have been observed in nanoscale, very often in an isolated experimental setup, and whether they reflect microscale properties of DNA. For this purpose, we studied the influence of DAU on the mechanics and size of the cell nucleus. We found that short-time treatment with DAU increased the stiffness of the isolated cell nucleus (Figure 5a). This indicates the capability of DAU to affect the structure of the nucleus and genome. An increase in stiffness can be a result of the disintegration of higher order structures and unfolding of DNA molecules. The treatment with 10 µM of DAU results in more than 10 times higher stiffness than in untreated cells. Additionally, we found that the size of the nucleus is also affected. The average area of the nucleus was decreased after the treatment with 10 µM of DAU. Both analyses indicate that previously observed disturbances of the DNA structure may have an impact on the microscale genome structure.

## 3. Materials and Methods

### 3.1. Materials

We used three anthracycline antibiotics: doxorubicin (DOX), daunorubicin (DAU) and epirubicin (EPI) (Sigma-Aldrich, Saint Louis, MI, USA). The techniques we used for the DNA studies required different lengths of DNA strands. Double stranded DNA (Oligo.pl, Warsaw, Poland) with the length of about 500 bp was used for the AFM studies. For the OT studies, we used longer DNA strands (4 kbp), modified at both ends with biotin (Oligo.pl, Warsaw, Poland). The properties of the used DNA strands for the AFM studies, were the following: (1) 35% GC—501 bp, (2) 52% GC—489 bp and (3) 77% GC—496 bp. For the OT studies, we used DNA: (1) 30% GC + biotin—4000 bp, (2) 50% GC + biotin—4000 bp and (3) 70% GC + biotin—4000 bp. The shortest DNA strands with the length of 40 bp were used in the molecular dynamics simulations. We used DNA with (1) 30% GC bp, (2) 50% GC bp and (3) 70% GC bp.

AML-N-007 is a primary acute myeloid leukaemia cell line. Cells were cultured in Advanced RPMI (Gibco, Thermo Fisher Scientific, Waltham, MA, USA) supplemented with 15% fetal calf serum (FCS, Gibco, Thermo Fisher Scientific, Waltham, MA, USA). For the cell nuclei isolation, Nuclei EZ lysis buffer (Sigma Aldrich, Saint Louis, MI, USA) was used. After isolation, cells’ nuclei were stored in Nuclei EZ storage buffer (Sigma Aldrich, Saint Louis, MI, USA) at −80 °C. 

### 3.2. The Structural Study of the Anthracycline Interactions with Double Helix of DNA

Examinations of the changes in the DNA structure under the influence of anthracycline antibiotics were carried out using atomic force microscopy.

#### 3.2.1. Preparation of Samples for Studying Structural Changes of DNA Using AFM

For AFM studies, commercial mica platelets (Plano GmbH, Wetzlar, Germany) were used as the subphase. Each DNA sample—35% GC, 52% GC and 77% GC—was dissolved in a buffer containing 150 mM MgCl_2_ (Sigma Aldrich, Saint Louis, MI, USA), 150 mM KCl (Sigma Aldrich, Saint Louis, MI, USA) and 20 mM of HEPES (Sigma Aldrich, Saint Louis, MI, USA), pH 7.4. The Mg^2+^ ions were needed for the deposition of DNA on the mica surface. Next, solutions of the studied cytostatic drugs in ultra-pure Mill-Q water were added to the DNA samples to obtain a final concentration of DNA of 3 nM and anthracyclines of 0.3 μM or 2 μM. A concentration of 0.3 μM provides anthracycline molecules in a sufficient amount to intercalate about 20% of the potential sites in the entire nucleotide sequence. In turn, a concentration of 2 μM provides an excess amount of drug molecules relative to the base pairs in the DNA strand. A control sample of DNA without drugs was also prepared. Subsequently, the samples were incubated for 3 h at room temperature. After the incubation, 10 μL of each sample was placed on the freshly cleaved mica surface. After 60 s, the mica surface was washed with Milli-Q water for several times and blown dry in a gentle stream of nitrogen gas. Finally, the mica was fixed with double-sided tape on a metal disk. The samples were placed in the AFM microscope and imaging was started.

#### 3.2.2. AFM Imaging Technique

The measurements were carried out using a Nanoscope IIId scanning probe microscope with Extender Module (Bruker, Billerica, MA, USA) in an air atmosphere at room temperature using the tapping mode. Silicon scanning probes with a resonance frequency in the range of 183–192 kHz, elastic constant of 43 N/m and tip diameter of 10 nm were used for the measurements. The set value of the probes’ vibration amplitude was maintained, by the feedback system, up to 80% of the free oscillation amplitude of the probe. The scanning frequency was between 0.500 and 1.500 Hz, the scanning angle was 0°.

#### 3.2.3. AFM Data Analysis

AFM images were processed in the Nanoscope program, version 6.13.1.0 (Veeco Instruments Inc., Plainview, NY, USA). By using the “flatten” and “plane-fit” functions, the noise was reduced and the plane of the images was leveled. Next, measurements of the length of the DNA strands (contour length) and particle heights based on cross-section profiles of the samples were carried out. Analysis was performed for at least 150 untangled DNA, placed entirely in the image area.

### 3.3. Changes in DNA Stiffness upon Treatment with Anthracycline Antibiotics

Examinations of the changes in DNA stiffness under the influence of anthracycline drugs were carried out using optical tweezers (OT).

#### 3.3.1. Preparation of Samples for Studying Mechanical Changes of DNA Using Optical Tweezers

DNA was dissolved in HEPES buffer to a concentration of 3 nM. The resulting DNA solution was separated into four Eppendorf tubes with a volume of 1.5 mL. Three samples of DNA were treated with cytostatics by adding a solution of DOX, DAU or EPI in Mili-Q water, so the final drug concentration was 12 μM, which corresponded to about a 100% base pair intercalation. The fourth tube contained a DNA control sample without a drug. Thereafter, a suspension of polystyrene beads and samples for the measurement with OT were prepared as follows. The suspension with beads contained 250 μL of HEPES buffer, 50 μL BSA (Sigma Aldrich, Saint Louis, MI, USA) (preventing balls from sticking together) and 15 μL of suspension containing 3.4 μm polystyrene beads coated with streptavidin (SPHERO™, Spherotech Inc., Lake Forest, IL, USA). The control sample consisted of 100 μL of HEPES buffer, 10 μL of suspension with beads and 10 μL of DNA solution in HEPES buffer. Tested samples contained 100 μL of HEPES buffer, 10 μL of suspension with beads and 10 μL of DNA solution in HEPES buffer with cytostatics.

The presence of streptavidin on the surface of the polystyrene beads enabled the attachment of DNA chains to them via the biotin molecules present at their ends. Biotinylated DNA chains were prepared by Oligo.pl (Warsaw, Poland) using a chemical biotinylation method. A DNA biotin labelling was attached to the 5′-ends of the DNA strands using a C6 standard spacer (DMT-Biotin-C6 Phosphoramidite, Metabion International AG, Planegg, Germany).

Then, the test and control samples were placed in the measuring chambers. A one-millimeter-high chamber was sandwiched between a microscope slide and a coverslip. The 0.5 mm thick chamber made of a sterile tape was filled with 23 µL of the prepared solution. The prepared samples were then placed in a measuring system. The measurements were performed for at least 20 dsDNA molecules in each sample.

#### 3.3.2. OT Technique

The preparations for stretching the DNA with optical tweezers started with mixing streptavidin-coated microbeads (diameter 3.05 µm, SPHERO™, Spherotech Inc., Lake Forest, IL, USA) with the solution of biotinylated DNA. When two microbeads are close enough, it often happens that a DNA chain binds with its one end to the first bead and with the other end to the second bead. As a result, the two beads get connected with each other by a DNA chain. While one bead is held fixed (i.e., it does not move), the second bead is pulled away from the first one. As the beads move further and further away from each other, the DNA chain between them starts to untangle until it gets entirely straightened to its actual length (in our case 4000 bp = 1.36 µm). Then, the stretching begins. The elongation of the chain is linearly proportional to the applied force in the Hooke regime. Then, the strain of the chain reaches its limit and the chain breaks. It is a very characteristic moment and can be easily observed both during the measurement under the microscope and later during the analysis of the beads’ trajectories.

In our experiments, the stretching is performed in a quasi-static manner. It means that the second bead is shifted away from the first one slowly, step by step. The entire stretching process lasts for 20–30 s. One should note that in the so-called low Reynolds number regime, for a microscopic bead moving through a relatively viscous medium, there is substantial drag acting upon the bead [30] and the higher the speed, the greater the drag. This unwanted factor may lead to overestimation of the stretching force, especially while shifting the bead rapidly. However, in the quasi-static approach, the drag on the bead is minimized due to the slow movement. Additionally, long time steps guarantee that there is enough time for the DNA chain to react with the exerted stretching force as well as for the holographic device (spatial light modulator) to set the exact position of the optical trap [31].

### 3.4. Cell Nuclei Isolation for Optical Tweezers Studies

AML-N-007 cells were suspended in 1 mL of DNR solution at concentrations of 1, 5 and 10 µM and incubated at 37 °C for 1 h. After incubation, cells were washed three times in PBS (phosphate-buffered saline solution, Ludwik Hirszfeld Institute of Immunology and Experimental Therapy, Wroclaw, Poland) supplemented with 2.5% FCS. Next, cells were suspended in 1 mL of Nuclei EZ lysis buffer for 5 min, washed once in PBS and suspended in lysis buffer for another 5 min. Isolated nuclei were washed three times in PBS and stored in Nuclei EZ storage buffer at −80 °C.

For OT studies, cell nuclei were biotinylated. We used EZ-link Sulfo-NHS-LC-Biotin (Thermo Fisher Scientific, Waltham, MA, USA) at a concentration of 1.5 mg/mL (2.69 mM). Nuclei were incubated at 4 °C for 30 min and then washed three times in PBS. OT samples were made by mixing 100 µL of cell nuclei suspended in PBS, 100 µL of BSA and 3 µL of aqueous solution of streptavidin-coated microbeads (diameter 4.5 µm, SPHERO™, Spherotech Inc., Lake Forest, IL, USA). A one-millimeter-high chamber was sandwiched between a microscope slide and a coverslip. The chamber made of sterile tape was filled with 23 µL of the prepared solution. 

### 3.5. Molecular Dynamics

To test the molecular effect of aminoglycoside antibiotics on the elongation of the 40 bp DNA strand as a function of the amount of GC pairs, the following numerical models with fully atomic description were prepared: (1) DNA model with 30% of GC pairs (CTTAATGCCTTAACATTGTGTATAACGAAAGGAAATATCT), (2) DNA model with 50% of GC pairs (CTATAAGGTGAAGGCTCATGGCAAGAAAGTGCTCGGTGCT) and (3) DNA model with 70% of GC pairs (CCCTAAGGTGCCGGCTCATGGCAAGCCCGTGCTCGGTGCC). Each model was tested in four different configurations: (A) without antibiotics, (B) with four molecules of DOX, (C) with four molecules of DAU and (D) with four molecules of EPI. DNA strand models were generated using the make-Na server [32], then hydrated up to the volume box of 15^3^ nm^3^ with 0.15 mol/L ionized NaCl within TIP3 water molecules. The obtained twelve numerical systems (1A–D, 2A–D, 3A–D) were first each equilibrated over 25 ns and then each simulated for a following 50 ns under constant number of particles, pressure and temperature (NPT) conditions, using the NAMD 2.9 molecular dynamics software [33], with 1 atm pressure and 300 K temperature parameters. Long-range electrostatic forces were taken into account using the particle mesh Ewald approach [34], and the integration time step was equal to 2 fs. The used CHARMM 36 force field for nucleic acids was completed with parameters for daunorubicin, doxorubicin and epirubicin, based on ab initio quantum mechanics calculations at the B3LYP/6–31+g level of theory, using the Gaussian 2016 suite of programs [35], with a procedure described elsewhere [17].

## 4. Conclusions

In our study, we demonstrated that anthracycline antibiotics intercalate and interact with the double helix of DNA, specifically between adjacent GC pairs, causing structural changes in dsDNA. The most notable consequence of such an interaction is DNA elongation and decrease in height. It is accompanied by a reduction in the double helix stiffness and force needed to break dsDNA. Thus, our analysis reveals greater susceptibility of anthracycline-treated DNA to chains break.

Moreover, we demonstrated also that isolated nuclei are able to respond to force by adjusting their stiffness to resist the applied tension. Isolated from AML cells, cellular nuclei were examined by using optical tweezers. Cellular mechanical changes in response to the chemotherapeutic agents were observed. The study was designed to compare the effects of three different anthracyclines on dsDNA with various ratios of GC pairs. Reassuming, it can be concluded that:
All anthracyclines affected the topology of the dsDNA molecule, lengthening and decreasing the height of the dsDNA molecule;The effect that anthracyclines exert on the dsDNA topology was dependent on the GC/AT ratio in dsDNA, and was strongest for the GC-rich dsDNA, suggesting preferential intercalation for GC pairs;Despite the high structural similarity of the DOX, DAU and EPI anthracyclines, they differ in the extent of their effect on the topology and their dependence on the composition of dsDNA. DOX exerts the strongest effect on dsDNA topology. DOX and DAU had a preference for GC-rich DNA, while EPI caused similar disturbances in GC-rich and balanced dsDNA topology;The malformations of the DNA topology were accompanied by a reduction in DNA stiffness and its susceptibility to break. The worsening of the biomechanical properties of dsDNA is dependent on the dsDNA composition and structure of the interacting molecule. The GC-rich dsDNA was especially affected by anthracyclines, once more indicating a preferential interaction with GC pairs. This is especially important because DNA regulatory elements are often located in GC-rich regions of genomes.

## Figures and Tables

**Figure 1 ijms-21-04142-f001:**
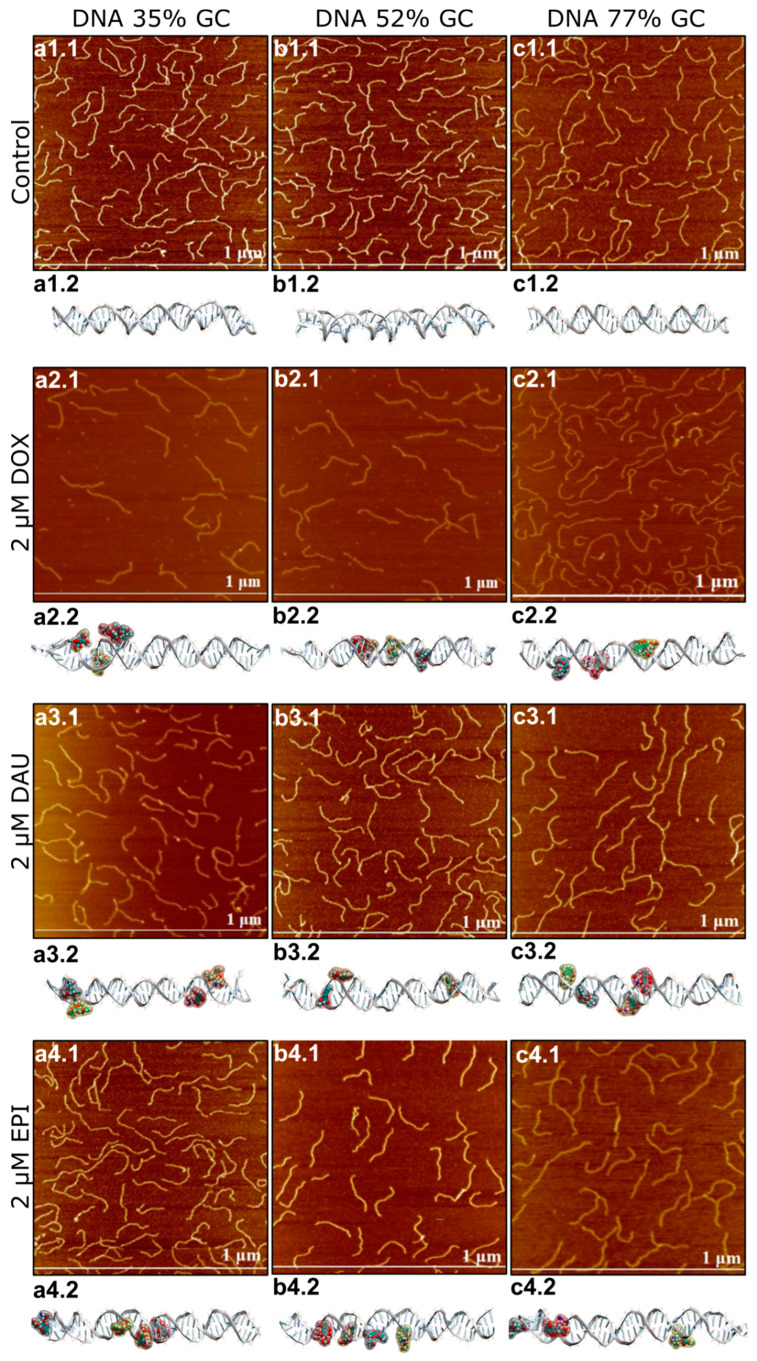
Atomic force microscopy (AFM) and molecular dynamics images of DNA be fore and after the interaction with anthracyclines in the concentration of 2 µM. For DNA 35% GC AFM images are shown in figures (**a1.1**–**a4.1**) and molecular dynamics images are shown in figures (**a1.2**–**a4.2**) For DNA 52% GC AFM images are shown in figures (**b1.1**–**b4.1**) and molecular dynamics images are shown in figures (**b1.2**–**b4.2**). For DNA 77% GC AFM images are shown in figures (**c1.1**–**c4.1**) and molecular dynamics images are shown in figures (**c1.2**–**c4.2**).

**Figure 2 ijms-21-04142-f002:**
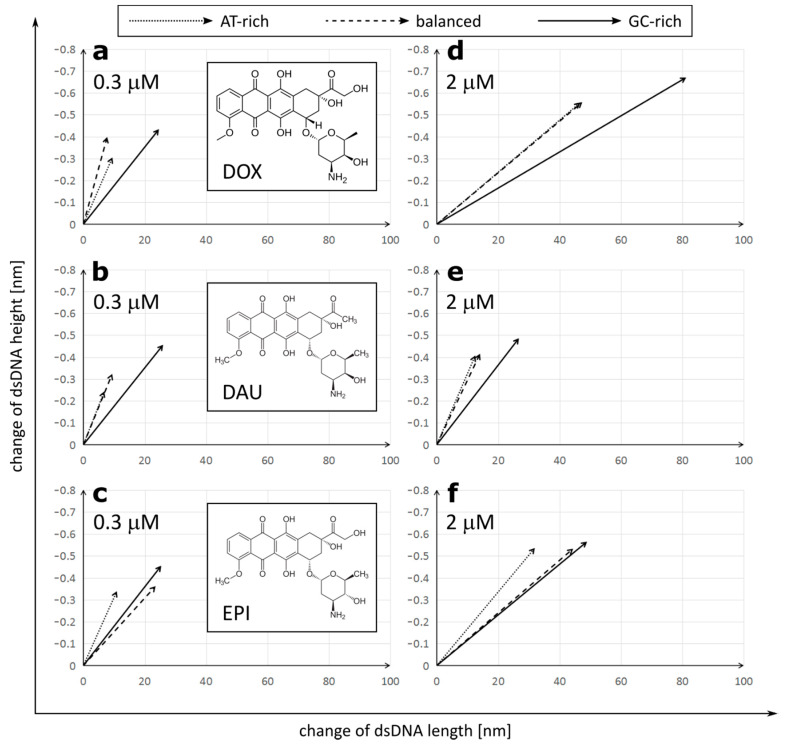
Correlation of dsDNA length and height changes depending on the anthracycline (**a**,**d**—doxorubicin (DOX); **b**,**e**—daunorubicin (DAU); **c**,**f**—epirubicin (EPI)) type and concentration (**a**,**b**,**c**—0.3 µM; **d**,**e**,**f**—2 µM). The raw data are included in Appendix A.

**Figure 3 ijms-21-04142-f003:**

The scheme of the stretching process of (**a**) DNA chain and (**b**) cell nucleus. Two microbeads are trapped in two separate optical traps and are connected with each other by the DNA or cell nucleus. One bead (left) is immobile, whereas the second bead (right) is slowly pulled away from the first one.

**Figure 4 ijms-21-04142-f004:**
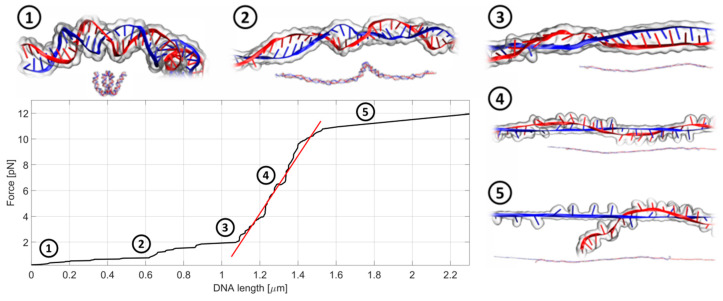
An exemplary record of the stretching experiment in a quasi-static manner. In the first stage, the DNA chain gets untangled and straightened to its actual length, dictated by the number of base pairs in the chain. This corresponds to the flat part of the curve (step 1 and 2). Next, the actual stretching and lengthening of the DNA chain occurs (step 3 and step 4) followed by the DNA breakage (step 5). The slope of the fitted linear function (red) is a direct measure of the chain stiffness. The whole process takes about 20–30 s.

**Figure 5 ijms-21-04142-f005:**
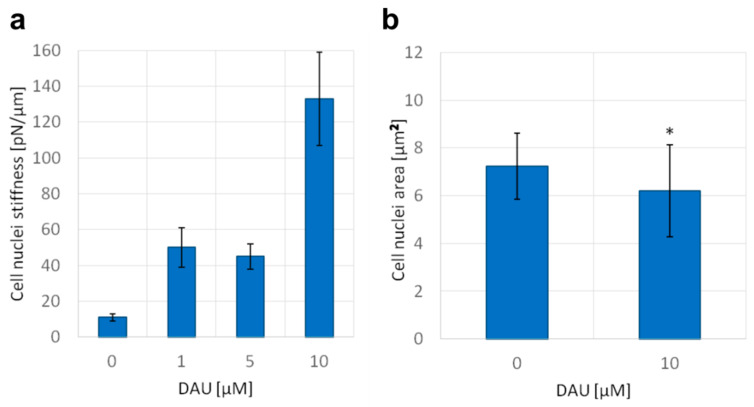
Daunorubicin affects the stiffness of isolated nuclei and the size of the nucleus of AML cells. (**a**) Nuclei stiffness, mean +/− standard deviations of the isolated nuclei are shown on the graph (*n* = 15), (**b**) area of nuclei of AML cells stained with PureBlu Hoechst 33342, average +/− standard deviations are shown on the graph (*n* = 40), asterisk indicates statistically significant difference (*p* < 0.01).

**Table 1 ijms-21-04142-t001:** DNA stiffness and breakage force. Doxorubicin (DOX), daunorubicin (DAU) and epirubicin (EPI) concentrations were 12 µM.

DNA Stiffness [pN/µm]			
Sample	AT-Rich	Balanced	GC-Rich
Untreated	24 ± 5	21 ± 4	22 ± 5
DOX	18 ± 5	14 ± 3	8 ± 3
DAU	12 ± 1	19 ± 3	11 ± 4
EPI	12 ± 1	15 ± 4	10 ± 3
**DNA Breakage Force [pN]**			
**Sample**	**AT-Rich**	**Balanced**	**GC-Rich**
Untreated	-	8.0 ± 1.5	12.6 ± 1.5
DOX	-	11.0 ± 2.5	9.0 ± 1.8
DAU	-	9.0 ± 1.5	9.6 ± 1.5
EPI	8.5 ± 4.5	11.0 ± 0.5	9.3 ± 1.8

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
