# Peer review of "Profound Nanoscale Structural and Biomechanical Changes in DNA Helix upon Treatment with Anthracycline Drugs"

_ijms, 2020, doi:10.3390/ijms21114142_

Round 1

Reviewer 1 Report

Profound nanoscale structural and biomechanical changes in DNA helix upon treatment with anthracycline drugs

The present manuscript describes the results of intercalation of different anthracycline antibiotics, namely doxorubicin (DOX), epirubicin (EPI) and idarubicin (IDA), in ds DNA. The results were obtained at nanoscale and single molecule level by employing Atomic Force Microscopy (AFM) and Optical Tweezer assays. The manuscript sheds light on the detrimental effects of all three-anthracycline antibiotics on ds DNA and their particular affinity towards GC pairs and GC-rich regions of DNA. The scientific results described in the study are interesting and the manuscript is well written and easy to follow.

However, I would like to suggest some minor revisions and additions to the manuscript before recommending it for publication.

  1. The authors can provide a small discussion comparing current results with those of similar studies on antibiotics such as PD153035 and Cisplatin, which were quoted by the authors in the manuscript. This will benefit the manuscript and make it viable for a wider audience.
  2. The authors should provide a bit more insight (either in the main text or in the supplementary information) into how they performed the linear fitting of the optical tweezer data.
  3. In the legend of Figure 4, the authors should elucidate upon the 5 stages showcased in the figure.
  4. Though the manuscript is well written, it is punctuated with minor grammatical and syntax errors. I mention two of the randomly selected ones;
  • Line 56 - Most of the nuclear activity of anthracycline antibiotics its attributed to its ability to…….
  • Line 66 - …..nucleotide sequence, although it was have been demonstrated that anthracycline antibiotics may…..

Overall, I will advise a thorough editing and revision of the manuscript

Author Response

Response to Reviewer 1 Comments

Dear Reviewer,

we are grateful for your comments and suggestions. We prepared revised manuscript, detailed revision. In attached revision we placed the answers to all your comments. Changes and additions to the text of the manuscript have been marked in red. English has been improved, changes have also been marked in red in the text.

The present manuscript describes the results of intercalation of different anthracycline antibiotics, namely doxorubicin (DOX), epirubicin (EPI) and idarubicin (IDA), in ds DNA. The results were obtained at nanoscale and single molecule level by employing Atomic Force Microscopy (AFM) and Optical Tweezer assays. The manuscript sheds light on the detrimental effects of all three-anthracycline antibiotics on ds DNA and their particular affinity towards GC pairs and GC-rich regions of DNA. The scientific results described in the study are interesting and the manuscript is well written and easy to follow.

However, I would like to suggest some minor revisions and additions to the manuscript before recommending it for publication.

The authors thank the reviewer very much for the careful evaluation of our work.

Point 1: The authors can provide a small discussion comparing current results with those of similar studies on antibiotics such as PD153035 and Cisplatin, which were quoted by the authors in the manuscript. This will benefit the manuscript and make it viable for a wider audience.

Response 1: The reviewer asked to remove this sentence, so we agree with the comment. As a result we remove this sentence from the introduction section, but add references to both cited works in “Results and discussion” section:

in chapter 2.2, line 152: “Interestingly the increment of DNA contour can be observed upon treatment with PD153035 in OT studies [26].”

in chapter 2.3, line 213: “This is consistent with previous observations [27], where the addition of other DNA interacting drug: cisplatin, resulted in decrease of persistent length with increased concentration of drug. Interestingly, cisplatin does not primarily intercalate to DNA, suggesting that all modes of interaction can substantially affect the homeostasis of dsDNA molecule.”

Point 2: The authors should provide a bit more insight (either in the main text or in the supplementary information) into how they performed the linear fitting of the optical tweezer data.

Response 2: Standard linear regression was used to perform the linear fitting. The software used was Curve Fitting Toolbox, MATLAB (Mathworks). A proper comment was added to the main text (chapter 2.3, line 189).

Point 3: In the legend of Figure 4, the authors should elucidate upon the 5 stages showcased in the figure.

Response 3: Thank you for pointing out this issue. Appropriate changes were made in the legend of Figure 4. The revised text articulates as follows:

“An exemplary record of the stretching experiment in a quasi-static manner. In the first stage, DNA chain gets untangled and straightened to its actual length dictated by the number of base pairs in the chain. This corresponds to the flat part of the curve (step 1 and 2). Next, the actual stretching and lengthening of DNA chain occurs (step 3 and step 4) followed by the DNA breakage (step 5). The slope of the fitted linear function (red) is a direct measure of chain stiffness. The whole process takes about 20–30 seconds.”

Point 4: Though the manuscript is well written, it is punctuated with minor grammatical and syntax errors. I mention two of the randomly selected ones;

  • Line 56 - Most of the nuclear activity of anthracycline antibiotics its attributed to its ability to…….
  • Line 66 - …..nucleotide sequence, although it was have been demonstrated that anthracycline antibiotics may…..

Point 5: Overall, I will advise a thorough editing and revision of the manuscript

Response 4&5: We are very obliged for the reviewer’s comment that helps to improve the manuscript, substantially. We have revised the manuscript, thoroughly and imperfections have been corrected, conscientiously.

Reviewer 2 Report

The manuscript by Kaczorowska et al. reports the AFM imaging and optical tweezers stretching studies of dsDNAs and the effects of three DNA intercalators (doxorubicin (DOX), epirubicin (EPI) and idarubicin (IDA)). In addition, the authors carried molecular dynamics simulation studies on how the small molecules bind to dsDNAs. I’m fine with the AFM and modelling studies. But the results from the optical tweezers studies are not consistent the well established dsDNA data. The main problem is that the obtained force value needed to break the dsDNA is way too low (~10 pN). The authors should show representative curves for each dsDNA at different conditions. The breaking force should be at least 65 pN. In addition, people typically characterize the stiffness of dsDNA by using persistent length through fitting a Worm-Like-Chain model. But the authors has not mentioned the persistent lengths in the manuscript.

Figure 2, The data for two concentrations are now shown in two panels. They can be combined in one panel.

Lines 98-101, remove the sentence.

Lines 151-163, GT rich should be GC rich.

Line 300, mention how the dsDNA is labelled with biotin.

Ref 26 is the same as ref 28.

Round 2

Reviewer 2 Report

I'm still not convinced that the DNA duplex can break at around 10 pN by stretching along the helical axis. Please check this paper (https://pubs.acs.org/doi/10.1021/jacs.8b10252 ), which shows that a DNA duplex is stable at 10 pN for >1000s. It is quite possible that the breaking forces the authors measured are not those of the DNA duplex.